# Physiological Implication of Slope Gradient during Incremental Running Test

**DOI:** 10.3390/ijerph191912210

**Published:** 2022-09-26

**Authors:** Johan Cassirame, Antoine Godin, Maxime Chamoux, Gregory Doucende, Laurent Mourot

**Affiliations:** 1Laboratory Culture Sport Health and Society (C3S−UR 4660), Sport and Performance Department, University of Bourgogne Franche-Comte, 25000 Besançon, France; 2EA7507, Laboratoire Performance, Santé, Métrologie, Société, 51100 Reims, France; 3Mtraining, R&D Division, 25480 Ecole-Valentin, France; 4EA3920-Prognostic Markers and Regulatory Factors of Heart and Vascular Diseases, and Exercise Performance, Health, Innovation Platform, University Bourgogne Franche-Comté, 25000 Besançon, France; 5Laboratoire Interdisciplinaire Performance Santé en Environnement de Montagne (LIPSEM), UR-4604, Université de Perpignan Via Domitia, 7 Avenue Pierre de Coubertin, 66120 Font-Romeu, France

**Keywords:** trail running, exercise physiology, maximal oxygen consumption, performance, testing

## Abstract

Uphill running induces a higher physiological demand than level conditions. Although many studies have investigated this locomotion from a psychological point of view, there is no clear position on the effects of the slope on the physiological variables during an incremental running test performed on a slope condition. The existing studies have heterogeneous designs with different populations or slopes and have reported unclear results. Some studies observed an increase in oxygen consumption, whereas it remained unaffected in others. The aim of this study is to investigate the effect of a slope on the oxygen consumption, breathing frequency, ventilation and heart rate during an incremental test performed on 0, 15, 25 and 40% gradient slopes by specialist trail runners. The values are compared at the first and second ventilatory threshold and exhaustion. A one-way repeated measures ANOVA, with a Bonferroni post-hoc analysis, was used to determine the effects of a slope gradient (0, 15, 25 and 40%) on the physiological variables. Our study shows that all the variables are not affected in same way by the slopes during the incremental test. The heart rate and breathing frequency did not differ from the level condition and all the slope gradients at the ventilatory thresholds or exhaustion. At the same time, the ventilation and oxygen consumption increased concomitantly with the slope (*p* < 0.001) in all positions. The post-hoc analysis highlighted that the ventilation significantly increased between each successive gradient (0 to 15%, 15% to 25% and 25% to 40%), while the oxygen consumption stopped increasing at the 25% gradient. Our results show that the 25 and 40% gradient slopes allow the specialist trail runners to reach the highest oxygen consumption level.

## 1. Introduction

The physiology of running has been largely investigated in the last century to explore human physiology and performance [1,2,3,4]. The bipedal locomotion speed is commonly used to prescribe an exercise intensity with the aim of analyzing a physiological request [5]. For the purposes of clinical and sports performance, incremental running tests have been developed to evaluate the maximal cardiorespiratory capabilities [6] to perform clinical diagnoses or prescribe physical training programs. Such evaluations have been designed mainly on treadmill machines, allowing an increase in the exercise intensity by adjusting the speed, slope or both [7,8]. Such testing procedures are massively used in sports science because they allow the subject to reach the maximum oxygen uptake ($\dot{V}O_{2}max$), which is an important contributor to performance in many sports [3,9,10]. These procedures also allow the determination of physiological landmarks, such as ventilatory thresholds, that allow the setting of intensity levels during a training program [11,12]. On a level condition, the protocols designed to assess the $\dot{V}O_{2}\max$ during running could differ, but may lead to similar $\dot{V}O_{2}max$ values [13,14,15] if the progressivity and starting intensity are appropriately set.

In recent decades, the increase in popularity of trail running races [16] has attracted interest in uphill and downhill running evaluations, including the $\dot{V}O_{2}\max$ [17,18,19]. Physiologically speaking, the running economy and energy cost of uphill running has been largely described [2,20,21,22]. However, few studies have investigated the specific effect of a slope on the maximal physiological values, such as the $\dot{V}O_{2}max,$ during incremental running tests. The existing literature provides contrasting or opposite results as the studies have been conducted with different populations and/or testing protocols. The studies using a constantly increasing slope, [23] or with slopes from +7% to +25% [24,25,26] did not reported an alteration in the $\;\dot{V}O_{2}max$ with a positive slope, whereas others pointed out an increase in the$\;\;\dot{V}O_{2}max$ with a slope of up to 35% [27,28,29]. Hence, based on the current literature, it is not clear if a slope gradient can induce a significant alteration in the cardiopulmonary variables, especially with well-adapted athletes, such as trail runners, or if an optimal slope can be identified to reach the highest cardiorespiratory involvement, without being impacted by peripheral limitations, such as a lack of muscular force. Specifically, with trail runners, Balducci et al.’s or Schöffl et al.’s studies compared the maximal cardiorespiratory performance while running at 12.5, 16, and 25% on a treadmill and in ecological field situations [24,26]. They reported no significant change in the $\dot{V}O_{2}max$ but a progressive increase in the ventilation ($\dot{V}E$) with the slope. Contrary to this result, Scheer et al. reported a larger$\dot{\;V}O_{2}max$ and blood lactate concentration post-exhaustion (3 min) [28]. However, the protocol used for this study included a concomitant speed and slope increment during testing (+0.5 km·h^−1^ and +1% per minute) and provided a final slope of only around a +10% gradient.

Hence, the aim of this study was to examine the effects of different gradient slopes (from level to +40%) on the cardiorespiratory variables reached at exhaustion and the ventilatory thresholds during maximum incremental tests in specialist trail runners. Based on the previous studies, and especially on the physiological limitations of the $\dot{V}O_{2}max\;$[30], we hypothesized that the steeper the slope, the higher the $\dot{\;V}O_{2}max,$ since a steeper slope will require a larger muscular mass to elevate the body mass [31,32]. However, we also hypothesized that this phenomenon would tend to a plateau, so that a further increase in the slope would not lead to an increase and could lead to a decrease in the $\dot{V}O_{2}$, due to the peripheral limitations.

## 2. Methodology

### 2.1. Participants

Fourteen young trail runners, free of injury in the last six months, were involved in this study, with a training volume (8.4 ± 3.2 h) in last two months: four females (age: 20.2 ± 2.8 years, size 1.73 ± 0.03 m, weight 61 ± 7.1 kg), and ten males (age: 20.6 ± 2.2 years, size 1.76 ± 0.04 m, weight 65.1 ± 5.2 kg). The measurement period took place in the second part of the racing season, with no competitions in last two weeks preceding the measurements. All the athletes have more than four years of active practice of trail running, and the training volumes for both genders are, respectively, 8.7 ± 3 h and 8.9 ± 2.5 h. All the participants were informed of the design and aim of the study and provided their written consent to participate in this study. The experiment was conducted in accordance with the Declaration of Helsinki and received the approval ID-RCB: 2019-A03012-55 from “COMITE DE PROTECTION DES PERSONNES SUD MEDITERRANEE IV”.

### 2.2. Experiential Design

All the athletes involved in this study performed, in random order, four incremental test sessions with different constant slopes. All the sequence possibilities (24 different randomizations) were assigned to the athletes by a draw, eliminating successive sequences to avoid each athlete performing a similar sequence. The tests were completed in a period of two weeks, respecting at least three days of rest after each assessment and avoiding other strenuous activities. The protocols were designed with 0%, 15%, 25% and 40% positive slopes in ecological field conditions. The level protocol was performed on a track and field loop, whereas the slope protocols were performed on a regular track with a constant slope in a ski resort. The level protocol starts at 8 km·h^−1^ and increases by 0.5 km·h^−1^ every minute [33]; the 15% slope protocol starts at 3.37 km·h^−1^ and increases by 0.41 km·h^−1^; the 25% slope protocol starts at 2 km·h^−1^ and increases by 0.34 km·h^−1^; and the 40% protocol starts at 1.35 km·h^−1^ and increases by 0.27 km·h^−1^ (Figure 1). The uphill protocols were designed, based on previous experiments, in order to reach similar test durations whatever the slope, in line with the current recommendations [34]. These protocols start at the same ascending speed (AS) of 500 m per hour and with increments of 50 m per hour for the 15% slope condition, and of 100 m per hour for the 25% and 40% conditions. The speed control was performed using an audio soundtrack read by a mobile MP3 player. For the level condition, the pacing was done by a soundtrack reading a sound every 20 m [33]. Regarding the uphill test, the fixed pacing was done every 15 s. To maintain the right speed, the athlete must be at the flag position when the signal sounds. The test was interrupted if athletes deviated more than 5 m difference from the appropriate position during two successive intervals.

### 2.3. Physiological Measurements

During the incremental testing, the physiological parameters were measured using a portable gas exchange system, the Metamax 3B-R2 (Cortex Biophysics, Leipzig, Germany), previously validated by Marcfarlane et al. [35]. This system was installed on participants with a vest in a thoracic position and carefully placed on the clavicula to permit free arm movement while running. An oronasal face mask (7450 series V2 (HansRudolph, Shawnee, KS, USA)) was adjusted on each participant to install a bi-directional digital turbine. This turbine measured the respiration flow and obtained the $\dot{V}E$ in L.min^−1^ and the breathing frequency (*BF*) in cycles.min^−1^. A short sample line tube (0.6 m) collected the inspired and expired air between the mask and turbine to measure the O_2_ and CO_2_ concentrations and calculate the O_2_ consumption ($\dot{V}O_{2}$, L·min^−1^) and CO_2_ output ($\dot{V}CO_{2}$ L·min^−1^). For each subject, the $\dot{V}O_{2}$ was normalized and expressed in mL·min^−1^·kg^−^^1^. The heart rate (HR) was collected by a thoracic belt strap, the Polar H7 (Polar Electro, Kemplele, Finland), and transmitted via Bluetooth Low Energy technology to the gas exchange measurement system. For each data collection, the tests were initialized and started from a computer using MetaSoft Studio© software 5.5.1, and the data were collected into the internal memory of the portable device. Then, all the data were downloaded with the software to be stored and analyzed. Before each test, the flow sensor was calibrated with a 3L syringe and the gas sensors were calibrated with ambient air and the reference gas (15% O_2_, 5% CO_2_), as recommended by the manufacturer.

For each individual test, the MetaSoft Studio© software determined automatically the maximal physiological values in the highest average of 30 s. During this period (MAX), the $\dot{V}O_{2max}$ in mL·min^−1^·kg^−1^, ventilation ($\dot{V}E_{max}$) in L·min^−1^, breathing frequency ($BF_{max}$), heart rate ($HR_{max}$) were calculated. An experienced examinator determined the positions of the ventilatory thresholds 1 (VT1) and 2 (VT2) using Wasserman and Beaver’s method [36] to allow the system to extract following parameters: $\dot{V}O_{2vt1}$, $\dot{V}E_{vt1}$, $BF_{vt1}$, $HR_{vt1}$ and $\dot{V}O_{2vt2}$, $\dot{V}E_{vt2}$, $BF_{vt2}$ and $HR_{vt2}$. The VT1 and VT2 positions were set using the graphical interface of the MetaSoft Studio© software 5.5.1 displaying the $\dot{V}E/\dot{V}O_{2}$ and $\dot{V}E/\dot{V}CO_{2}$ curves over the time, with averaging every 10 s. The VT1 was set at the first increase of $\dot{V}E/\dot{V}O_{2}$ without an increase of $\dot{V}E/\dot{V}CO_{2}$, and the VT2 was set at a concomitant increase of $\dot{V}E/\dot{V}O_{2}$ and $\dot{V}E/\dot{V}CO_{2}$.

For each tests series with a gradient (15, 25 and 40%), the ascending speed in meters per hour was calculated and identified at the VT1, VT2 and the maximum moments.

### 2.4. Statistic

All the data exported from the MetaSoft Studio© software were merged into a Microsoft Office 365 Excel spreadsheet (Microsoft, Redmond, WA, USA) and computed to be analyzed with a custom R-Studio algorithm on the desktop software version 1.4.1106 (RStudio PBC, Boston, MA, USA). The descriptive statistics are presented as the mean and SD for the physiological variables and AS. The normal distribution of all the physiological variables was confirmed through the Shapiro–Wilk test (*p* > 0.05) except for the $\dot{V}O_{2}$, HR and $\dot{V}E$ measurements at the maximum moment on a 0% of slope. A one-way repeated measures ANOVA with a Bonferroni post-hoc analysis was used to determine the effects of the slope gradient (0, 15, 25 and 40%) on the physiological variables (HR, BF, $\dot{V}E$ and $\dot{V}O_{2}$) at specific time points: the VT1, VT2 and the maximum (MAX). In a second time, similar procedures were performed for the AS values for the gradient slopes of 15, 25 and 40% for each moment. The physiological variables not having a normal distribution were analyzed using the Friedman test to determine the impact of the slope gradient [37]. A post-hoc test was a paired Wilcoxon signed-rank test with the Bonferroni correction [38].

These analyses were complemented by an effect size estimation using Hedges’ g (population <16, repeated measures design) [39]. Hedges’ g was also used for the variables not following a normal distribution. The magnitude thresholds for the effect size were defined as 0.20, 0.60, 1.20, 2.0 and 4.0 for small, moderate, large, very large and extremely large correlation coefficients, in accordance with previous recommendations [40].

For this study a p level inferior at 0.05 was considered as significant.

## 3. Results

The durations of the tests were 944.0 ± 115.8 s, 907.6 ± 99.6 s, 900.2 ± 100.2 s and 904.3 ± 101.2 s for the 0, 15, 25 and 40% gradient slopes, respectively, without significant differences between the conditions. The data for each slope and time points (VT1, VT2 and MAX) are displayed in Table 1. All the individual values, as well as the mean and standard deviations are displayed as violin plot graphics in Figure 2 to observe the distribution and changes with the slope conditions. A second violin plots series was designed to show the AS values for the gradient conditions (Figure 3).

The results of the statistical analysis from the repeated one-way ANOVA and Bonferroni post-hoc analysis are displayed in Table 2 for all the physiological variables, whereas the ascending speed can be read in Table 3. At the VT1, VT2 and MAX time points, the $\dot{V}E$, $\dot{V}O_{2}$ and AS significantly increased with an increase in the slope gradients (*p* < 0.001), whereas the *HR* and *BF* were not significantly affected in any condition. Specifically, only the *HR* (at VT1; significantly increased with slopes; *p =* 0.006) and *BF* (at MAX; significantly increased with slopes; *p =* 0.011) were affected. Moreover, the AS showed significantly slower values (*p* < 0.001) between 15% and the other gradients (25 and 40%) but no difference was observed at any time points between the 25 and 40% slope gradients (*p* = 0.999).

## 4. Discussion

The aim of this study was to investigate the effect of a positive gradient slope on cardiopulmonary variables during maximal incremental testing. As hypothesized, we observed that the steeper the slope, the higher the$\dot{\;V}O_{2}max$ up to +25%, without a further significant increase thereafter. However, we also noticed that not all cardiorespiratory variables are equally influenced by the slope gradient increase.

As expected, our results confirmed an increase in the cardiorespiratory requirements while increasing the slope angles. However, not all cardiorespiratory variables were impacted equally. Indeed, we observed that the HR at VT1, VT2 and MAX remained unaffected by the slope gradients from 15% to 40%. This finding at MAX is in line with the previous studies [24,26,28,29]. This observation provides an interesting confirmation for training purposes and allows the transposition of the HR intensities in the various positive slopes when targeting the ventilatory threshold intensity based on the HR values.

Beyond the HR, the breathing pattern was impacted by the slope gradients. For the BF, we noted that no difference was found at the VT1 and VT2 landmarks. However, a weak trend was observed at the MAX, highlighting that the greater the slope, the higher the BF (*p* = 0.011) associated with a small effect size between the level condition and the +25% and +40% gradient or +15% and the steeper slopes. Moreover, we can observe in Figure 2 that the individual BF values are more dispersed for the 25 and 40% slopes. This observation is also corroborated with an increased standard deviation, especially at the MAX and for 40%. This bigger dispersion of the BF at the 25 and 40% gradients could be explained by different running pattern strategies when the slope increases. Indeed, runners can choose to increase cadence and decrease step-length, or vice-versa, [24] and it is known that a tight locomotor–respiratory coupling exists, especially while running [41]. Moreover this coupling could be exacerbated when the upper limbs are more involved [42,43] potentially, or in a more pronounced way, at higher gradients.

On the other hand, the $\dot{V}E$ was largely positively influenced by the increase in the slope for the VT1, VT2 and MAX time points. We observed small to moderate effect sizes between the conditions, excepting between +25% and 40% at the VT2 and MAX landmarks, which remained trivial. These results differ from the literature, since Balducci et al. did not observe any differences during the incremental running test for the level condition vs. the 15% and the level vs. the 25% slope gradients [24], and Scheer et al. did not report any differences between the level test and the test with the constant slope increase (1% per minute) [28]. Given that the BF remained largely unchanged, this phenomenon can only be explained by an increase in the tidal volume with an increasing slope to increase the air volume exchange at each breathing cycle. Similar finding have been reported by Lemire et al., comparing uphill and downhill to an incremental level running test [18]. In the same way, these authors also observed a higher metabolic coupling of ventilation when running uphill at 15% [18]. Moreover, the$\;\;\dot{V}E$ and tidal volume have already been demonstrated to be higher at the maximal intensity (>80% maximum) during an incremental test with a slope gradient compared to the level condition [27], or in a field test performed at 16% compared with the treadmill test at 1% [26]. Here again, the hypothesis of locomotor–respiratory coupling could potentially explain the mechanisms. Potentially, the slope gradient led to a reduction of the step frequency [24,44] at a similar intensity and could allow the use of a deeper breathing pattern with increased tidal volume.

Finally, we observed that the $\dot{V}O_{2}$ was higher during uphill running compared to the slope gradients for the VT1, VT2 and MAX time points. Amongst the different positive slopes, we noticed that, at VT1, no significant differences were found between the 15% and 25% (*p* = 0.201) and between the 25% and 40% gradients (*p* = 0.999). Both the VT2 and MAX time points provided the same pattern, with a significant increase in the $\dot{V}O_{2}\;$between 15% and 25% (*p* < 0.001 with a small effect size), while no differences were found between 25% and 40% (*p* = 0.739 and 0.999) These results corroborated the previous findings comparing the $\dot{V}O_{2}max$ between the level condition and the various slopes conditions [27,28,29,31,45,46]. Nevertheless, it is important to notice that these previous studies focused on a smaller slope gradient (7%). Moreover, differences in the studied populations could be reported as the previous studies were conducted without specialist trail runners [25] or with older and heavier athletes [24,26]. In these studies, the athletes’ ages were 38.5 ± 6.4 years and 42.8 ± 14.6 years, respectively, whereas the were 20.4 ± 2.3 years in our study, and the body weights were 69.8 ± 8.6 kg and 75.8 ± 10.2 kg vs. 63.9 ± 5.8 kg in our study.

The main mechanism that could explain an increase in the $\dot{V}O_{2}$ while running uphill is likely the higher muscular mass involved in this condition, especially for the vastus, soleus, gluteus, biceps femoris and gastrocnemius muscles [31,32,47,48,49]. In addition, uphill running over a 15% gradient eliminates the bouncing mechanism and the use of elastic energy, which is helpful for displacement [22,50,51]. The locomotion related to uphill running induces stronger muscular contractions and mechanical works [52]. To produce this additional mechanical energy, Robert and Belliveau concluded the involvement of a greater contribution of ankle, knee and hip extensor muscles in line with Sloniger et al. or Swanson and Caldwell’s work [31,32]. In addition, a greater hip extensors’ contribution has an additional effect on the metabolic expenditure, because theses muscles are not able to produce force economically with a low contribution of elastic storage and recovery [53,54]. Gottschall and Kram reported an increase of 75% in the propulsive force during treadmill running at a 9% slope compared to a level condition [55]. Basically, more important propulsive work induces additional concentric muscle contraction levels, which are most costly physiologically [21].

Furthermore, even if our result indicates a positive influence of the slope on the $\dot{V}O_{2}$ at the VT1, VT2 and MAX time points, we can also underline that the 40% gradient does not produce significantly greater $\dot{V}O_{2\;max}$ values compared to the 25%, while the $\dot{V}E_{max}$ is superior on this gradient. Based on this finding, we can exclude ventilatory limitation in this condition to produce a higher oxygen consumption. We can hypothesis a higher O_2_ extraction, transportation or utilization in muscles as limiting factors at steep grades [30]. In addition, this result provides an interesting insight to evaluate trail runner specialists. Indeed, based on this result, we can recommend the evaluation of the $\dot{V}O_{2max}$ capacity of trail runners on slope conditions at or over 25% to obtain their real maximal capacities, which has been highly correlated with performance levels for short and middle-distance trail-running competitions [56].

Finally, our study investigated the maximal AS for the VT1, VT2 and MAX time points and for different slope conditions (15, 25 and 40%). Our results provided similar findings for all the time points; the 15% gradient provided a systematically lower AS than 25% (*p* < 0.001) and 40% (*p* < 0.001), with a very large effect size. We also noted that the AS did not differ between 25 and 40%, whatever the time points considered (*p* = 0.999). This observation is very interesting for training purposes and may allow using ascending speeds as intensity indexes during sessions. Moreover, this information provides new evidence for athletes intending to compete in maximal elevation challenges, where the goal is to reach the highest elevation in an allocated duration (from 4 to 24 h). In our study, the slopes between 25 and 40% allowed the achievement of a greater ascending speed (approx. 1750 m per hour) than the 15% gradient (approx. 1175 m per hour) and could be selected for such contests based on the individual’s capabilities.

## 5. Conclusions

The aim of this study was to investigate the effect of different gradient slopes (from level to +40%) on the cardiorespiratory variables reached at exhaustion and ventilatory thresholds during maximum incremental tests in specialist trail runners. First, our study clearly indicates that slope conditions over a 15% gradient allow reaching higher $\dot{V}O_{2}$ and $\dot{V}E$ levels at the VT1, VT2 and MAX time points, whereas the BF and HR remain unchanged for the specialist trail runners. Our study also pointed out a limitation of the slope influence on the physiological variables over the 25% gradient, whereas the $\dot{V}O_{2}$ does not further increase at the VT1, VT2 and MAX time points. Finally, an analysis of the speed provides similar findings. Our study pointed out that the maximal AS can be reached for slopes of 25% and 40% equally. Based on these results, we recommend trail runner testing on slope conditions between 25 and 40% to stress and reach their real maximal cardiorespiratory capacities and obtain their maximal AS for training purposes.

## Figures and Tables

**Figure 1 ijerph-19-12210-f001:**
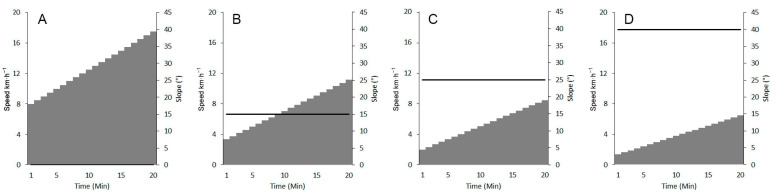
Graphical representation of each protocol performed by participants. The grey area represents the speed in km·h, while the black solid line represents the slope gradient in degrees. Protocol designs are displayed from 0° to 40° from left to right. ((**A**): 0°, (**B**): 15°, (**C**): 25° and (**D**): 40°).

**Figure 2 ijerph-19-12210-f002:**
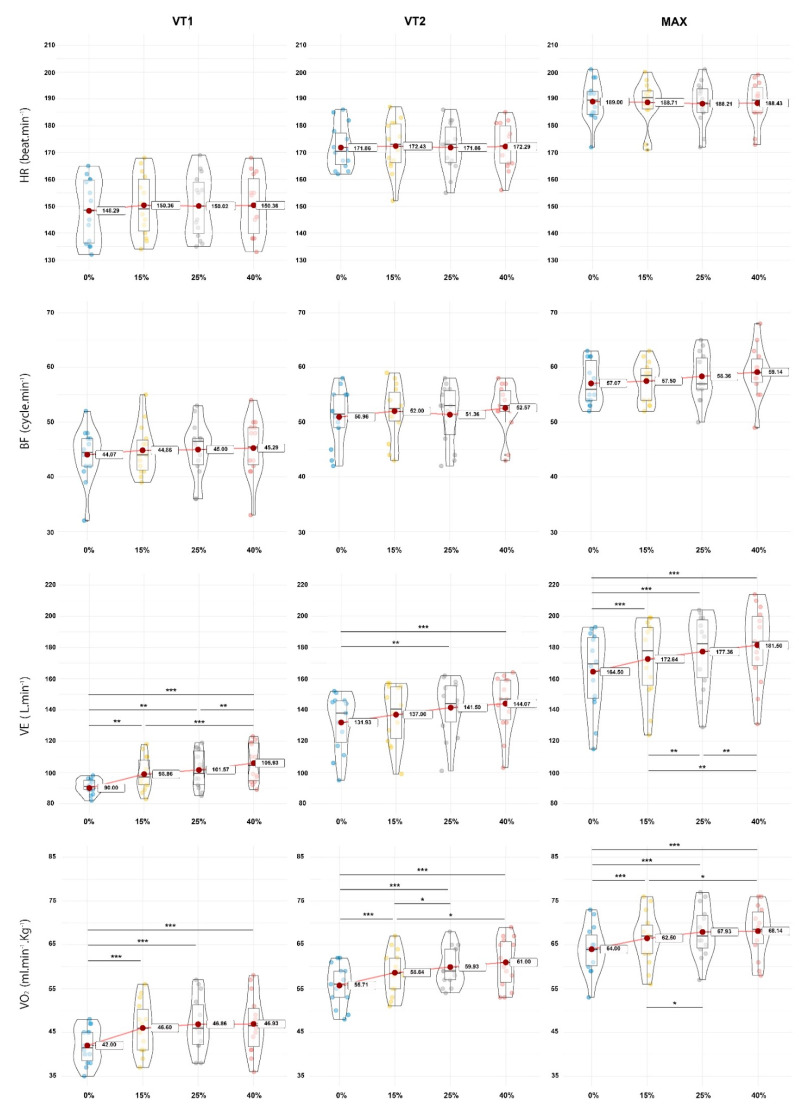
Violin plot of dependent variables, heart rate (HR), breathing frequency (BF), ventilation ($\dot{V}E$), oxygen consumption ($\dot{V}O_{2}$) for all time points (VT1, VT2 and MAX) for slope conditions (0, 15, 25 and 40%). Statistical differences observed were noted as follow; * *p* < 0.05, ** *p* < 0.01 and *** *p* < 0.001.

**Figure 3 ijerph-19-12210-f003:**
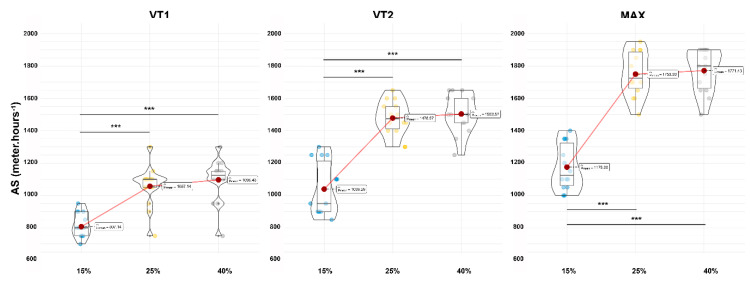
Violin plot of ascending speed (AS) at all positions (VT1, VT2 and MAX) for all slope conditions (15, 25 and 40%). Statistical differences observed were noted as follow; * *p* < 0.05, ** *p* < 0.01 and *** *p* < 0.001.

**Table 1 ijerph-19-12210-t001:** Mean and standard deviation of heart rate (HR), breathing frequency (BF), ventilation ($\dot{V}E$), oxygen consumption ($\dot{V}O_{2}$) and ascending speed (AS) for each position; ventilatory thresholds (VT1 and VT2) and at maximum (MAX) for 0, 15, 25, 40% gradient slope conditions. §, Ø and β indicate that value is significantly different than 0%, 15% and 25% gradient slope condition, respectively.

		0%	15%	25%	40%
**VT1**	HR (bpm)	148.3 ± 11.9	150.4 ± 11.4	150.1 ± 11.6	150.4 ± 11.4
	BF (cycles·min^−1^)	44.1 ± 4.8	44.9 ± 4.5	45 ± 5.1	45.3 ± 5.4
	VE (L·min^−1^)	90 ± 5.8 Ø β	98.9 ± 10.8 §	101.6 ± 11.9 §	105.9 ± 12.4 § Ø β
	VO_2_ (mL·min^−1^·kg)	42 ± 4.2 Ø β	46 ± 6 §	46.9 ± 6.2 §	46.9 ± 6.7 §
	AS (m·h^−1^)	-	807.1 ± 85.1 β	1057.1 ± 126.8	1096.4 ± 135.1 Ø
**VT2**	HR (bpm)	171.9 ± 8.2	172.4 ± 9.7	171.9 ± 9.1	172.3 ± 8.5
	BF (cycles·min^−1^)	50.9 ± 5	52 ± 5	51.4 ± 5.4	52.6 ± 4.5
	VE (L·min^−1^)	131.9 ± 18.2 β	137 ± 18.8	141.5 ± 17.9 §	144.1 ± 18.1 §
	VO_2_ (mL·min^−1^·kg)	55.1 ± 6 Ø β	58.6 ± 5 § β	59.9 ± 4.4 §	61 ± 5.6 § Ø
	AS (m·h^−1^)	-	1039.2 ± 163.1 β	1478.5 ± 106.9	1503.5 ± 120 Ø
**Max**	HR (bpm)	189 ± 7.6	188.7 ± 8.2	188.2 ± 8.1	188.4 ± 7.8
	BF (cycles·min^−1^)	57.1 ± 3.9	57.5 ± 3.6	58.4 ± 4.4	59.1 ± 4.7
	VE (L·min^−1^)	164.5 ± 25.3 Ø β	172.6 ± 24.8 §	177.4 ± 23.6 § Ø	181.5 ± 24.5 § Ø β
	VO_2_ (mL·min^−1^·Kg)	63.3 ± 7.3 Ø β	66.5 ± 5.9 § β	67.9 ± 5.8 § Ø	68.1 ± 5.9 § Ø
	AS (m·h^−1^)	-	1175 ± 141.1 β	1750 ± 137.2	1771.4 ± 132.6 Ø

**Table 2 ijerph-19-12210-t002:** Results of statistical analysis including Shapiro–Wilk test for normality and *p* value for one-way ANOVA and Bonferroni post-hoc analysis comparing heart rate (HR), breathing frequency (BF), ventilation ($\dot{V}E)$ and oxygen consumption ($\dot{V}O_{2}$) at all positions (VT1, VT2 and MAX) and between slope conditions. Effect sizes are presented in right part of the table using Hedges’ g method.

Anova One-Way Repeated Measures						Bonferroni Post-Hoc				Effect Size			
**HR**	**VT1**	** *Slope* **	** *p value* **	** *Normality* **	** *p value* **		** *p value* **			** *g Hedges* **			
		0%	0.074	Valid	0.006		**0%**	**15%**	**25%**		**0%**	**15%**	**25%**
		15%	0.449	Valid		**15%**	0.22			**15%**	0.172		
		25%	0.202	Valid		**25%**	0.13	0.999		**25%**	0.142	0.024	
		40%	0.352	Valid		**40%**	0.16	0.999	0.999	**40%**	0.173	0	0.241
	**VT2**	** *Slope* **	** *p value* **	** *Normality* **	** *p value* **		** *p value* **			** *g Hedges* **			
		0%	0.182	Valid	0.920		**0%**	**15%**	**25%**		**0%**	**15%**	**25%**
		15%	0.753	Valid		**15%**	0.999			**15%**	0.061		
		25%	0.806	Valid		**25%**	0.999	0.999		**25%**	0.001	0.059	
		40%	0.771	Valid		**40%**	0.999	0.999	0.999	**40%**	0.049	0.015	0.047
	**MAX**	** *Slope* **	** *p value* **	** *Normality* **	** *p value* **		** *p value* **			** *g Hedges* **			
		0%	0.453	Failed	0.899		**0%**	**15%**	**25%**		**0%**	**15%**	**25%**
		15%	0.068	Valid		**15%**	0.999			**15%**	0.035		
		25%	0.72	Valid		**25%**	0.999	0.999		**25%**	0.096	0.059	
		40%	0.287	Valid		**40%**	0.999	0.999	0.999	**40%**	0.074	0.034	0.026
**BF**	**VT1**	** *Slope* **	** *p value* **	*Normality*	** *p value* **		** *p value* **			** *g Hedges* **			
		0%	0.377	Valid	0.421		**0%**	**15%**	**25%**		**0%**	**15%**	**25%**
		15%	0.434	Valid		**15%**	0.999			**15%**	0.163		
		25%	0.444	Valid		**25%**	0.999	0.999		**25%**	0.181	0.028	
		40%	0.3	Valid		**40%**	0.5	0.5	0.999	**40%**	0.23	0.083	0.053
	**VT2**	** *Slope* **	** *p value* **	** *Normality* **	** *p value* **		** *p value* **			** *g Hedges* **			
		0%	0.453	Valid	0.182		**0%**	**15%**	**25%**		**0%**	**15%**	**25%**
		15%	0.49	Valid		**15%**	0.999			**15%**	0.209		
		25%	0.107	Valid		**25%**	0.999	0.999		**25%**	0.08	0.121	
		40%	0.054	Valid		**40%**	0.19	0.999	0.78	**40%**	0.336	0.117	0.329
	**MAX**	** *Slope* **	** *p value* **	** *Normality* **	** *p value* **		** *p value* **			** *g Hedges* **			
		0%	0.08	Valid	0.011		**0%**	**15%**	**25%**		**0%**	**15%**	**25%**
		15%	0.187	Valid		**15%**	0.999			**15%**	0.11		
		25%	0.586	Valid		**25%**	0.058	0.666		**25%**	0.301	0.207	
		40%	0.913	Valid		**40%**	0.051	0.064	0.307	**40%**	0.468	0.382	0.169
**VE**	**VT1**	** *Slope* **	** *p value* **	** *Normality* **	** *p value* **		** *p value* **			** *g Hedges* **			
		0%	0.643	Valid	<0.001		**0%**	**15%**	**25%**		**0%**	**15%**	**25%**
		15%	0.49	Valid		**15%**	0.01			**15%**	0.994		
		25%	0.163	Valid		**25%**	0.008	0.439		**25%**	1.2	0.232	
		40%	0.056	Valid		**40%**	<0.001	<0.001	0.002	**40%**	1.6	0.591	0.348
	**VT2**	** *Slope* **	** *p value* **	** *Normality* **	** *p value* **		** *p value* **			** *g Hedges* **			
		0%	0.095	Valid	< 0.001		**0%**	**15%**	**25%**		**0%**	**15%**	**25%**
		15%	0.118	Valid		**15%**	0.569			**15%**	0.266		
		25%	0.208	Valid		**25%**	<0.001	0.205		**25%**	0.515	0.238	
		40%	0.119	Valid		**40%**	<0.001	0.086	0.097	**40%**	0.651	0.372	0.139
	**MAX**	** *Slope* **	** *p value* **	** *Normality* **	** *p value* **		** *p value* **			** *g Hedges* **			
		0%	0.186	Failed	<0.001		**0%**	**15%**	**25%**		**0%**	**15%**	**25%**
		15%	0.097	Valid		**15%**	<0.001			**15%**	0.316		
		25%	0.193	Valid		**25%**	<0.001	0.01		**25%**	0.511	0.189	
		40%	0.689	Valid		**40%**	<0.001	0.003	0.036	**40%**	0.663	0.348	0.167
**VO_2_**	**VT1**	** *Slope* **	** *p value* **	** *Normality* **	** *p value* **		** *p value* **			** *g Hedges* **			
		0%	0.398	Valid	<0.001		**0%**	**15%**	**25%**		**0%**	**15%**	**25%**
		15%	0.614	Valid		**15%**	<0.001			**15%**	0.754		
		25%	0.409	Valid		**25%**	<0.001	0.201		**25%**	0.886	0.136	
		40%	0.848	Valid		**40%**	<0.001	0.505	0.999	**40%**	0.86	0.143	0.010
	**VT2**	** *Slope* **	** *p value* **	** *Normality* **	** *p value* **		** *p value* **			** *g Hedges* **			
		0%	0.22	Valid	<0.001		**0%**	**15%**	**25%**		**0%**	**15%**	**25%**
		15%	0.81	Valid		**15%**	<0.001			**15%**	0.588		
		25%	0.184	Valid		**25%**	<0.001	0.048		**25%**	0.898	0.264	
		40%	0.204	Valid		**40%**	<0.001	0.001	0.739	**40%**	0.994	0.43	0.205
	**MAX**	** *Slope* **	** *p value* **	** *Normality* **	** *p value* **		** *p value* **			** *g Hedges* **			
		0%	0.034	Failed	<0.001		**0%**	**15%**	**25%**		**0%**	**15%**	**25%**
		15%	0.749	Valid		**15%**	<0.001			**15%**	0.427		
		25%	0.889	Valid		**25%**	<0.001	0.042		**25%**	0.680	0.237	
		40%	0.482	Valid		**40%**	<0.001	0.039	0.999	**40%**	0.712	0.271	0.035

**Table 3 ijerph-19-12210-t003:** Results of statistical analysis including Shapiro–Wilk test for normality and *p* value for one-way ANOVA and Bonferroni post-hoc analysis comparing ascending speed (AS) at all positions (VT1, VT2 and MAX) and between slope conditions 15, 25 and 40%. Effect sizes are presented in right part of the table using Hedges’ g method.

Anova One-Way Repeated Measures						Bonferroni Post-Hoc			Effect Size		
**AS**	**VT1**	** *Slope* **	** *p value* **	** *Normality* **	** *p value* **		** *p value* **		** *g Hedges* **		
		15%	0.655	Valid	<0.001		**15%**	**25%**		**15%**	**25%**
		25%	0.046	Valid		**25%**	<0.001		25%	2.25	
		40%	0.036	Valid		**40%**	<0.001	0.999	40%	2.49	0.291
	**VT2**	** *Slope* **	** *p value* **	** *Normality* **	** *p value* **	** *p value* **			** *g Hedges* **		
		15%	0.013	Valid	<0.001		**15%**	**25%**		**15%**	**25%**
		25%	0.65	Valid		**25%**	<0.001		25%	3.09	
		40%	0.275	Valid		**40%**	<0.001	0.999	40%	3.15	0.214
	**Max**	** *Slope* **	** *p value* **	** *Normality* **	** *p value* **		** *p value* **		** *g Hedges* **		
		15%	0.094	Valid	<0.001		**15%**	**25%**		**15%**	**25%**
		25%	0.518	Valid		**25%**	<0.001		25%	4.01	
		40%	0.056	Valid		**40%**	<0.001	0.999	40%	4.23	0.154

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
