# Peer review of "Physiological Implication of Slope Gradient during Incremental Running Test"

_ijerph, 2022, doi:10.3390/ijerph191912210_

Round 1
Reviewer 1 Report (Previous Reviewer 3)
Accept in present form
Author Response
We thank the reviewer for his/her positive feedback on our manuscript
Reviewer 2 Report (Previous Reviewer 1)
The main aim of the study „ Physiological Implication of Slope Gradient during Incremental Running Test“ is to examine the effects of different gradient slopes (from level to +40%) on cardiorespiratory variables reach at exhaustion and ventilatory thresholds during maximum incremental tests in trail runner specialists.
The study is interesting. I would like to appretiate the efforts of the authors. However, some information is missing in the article, some facts need to be explained and add:
In what part of the sports season (racing season, preparatory, transitional, …) did the testing take place?
Although they are endurance-trained individuals, and although the length of the load was comparable for all gradients, the length of the test is at the upper limit of the recommended length. The length of the test, about 15 minutes, could affect the maximum ventilation values found. I recommend that you think about this comment, or mention it in your work.
I recommend to describe (mark with a number or letter) the individual pictures in Figure 1.
In Tables 1 and 2, I recommend to edit the VO2 text using the subscript.
Randomization of probands – can you describe in more detail the methodology and procedure of the draw?
How exactly was VT1 and VT2 determined? Describe the methodology of the determination.
You indicate that there was probably a change in tidal volume (VT). Why was this parameter not evaluated?
For clarity of the text and quicker understanding of the results, I recommend to briefly describe in the results section the results found in the tables.
How was the running speed controlled and how was its exact compliance controlled? I recommend the mechanism of implementation, management and control of the pace.
Line 123: “(60-cm)” – is it correct?
I recommend that the wording of the conclusion be reformulated so as not to repeat the literal definition of the project goal.
Author Response
Dear,
Thank you for your review and comments. You can find below our responses and the corresponding amendments added in the text (and highlighted in yellow).
The study is interesting. I would like to appreciate the efforts of the authors. However, some information is missing in the article, some facts need to be explained and add:
In what part of the sports season (racing season, preparatory, transitional, …) did the testing take place?
We added further information about it lines 87-88.
Although they are endurance-trained individuals, and although the length of the load was comparable for all gradients, the length of the test is at the upper limit of the recommended length. The length of the test, about 15 minutes, could affect the maximum ventilation values found. I recommend that you think about this comment, or mention it in your work.
This point open the discussion on bigger debate and depending on the “dogma” you are believing. Midgley et al. confirmed that such test on treadmill can be performed with exhaustion range from 5 to 26 minutes. On our study, the average duration of tests is around 900 seconds (15 minutes) as mentioned in the Results section (line 166). This duration is in the middle range of durations reported in the literature. We did not find any evidence that time to exhaustion during incremental test impact ventilation values.
Midgley AW, Bentley DJ, Luttikholt H, McNaughton LR, Millet GP. Challenging a dogma of exercise physiology: does an incremental exercise test for valid VO 2 max determination really need to last between 8 and 12 minutes? Sports Med. 2008;38(6):441-7. doi: 10.2165/00007256-200838060-00001. PMID: 18489192.
I recommend to describe (mark with a number or letter) the individual pictures in Figure 1.
Thank you for this input. We added letter as suggested and amend figure description
In Tables 1 and 2, I recommend to edit the VO2 text using the subscript.
Well noted, we amended VO2 based on your comment
Randomization of probands – can you describe in more detail the methodology and procedure of the draw?
Protocol contains 4 conditions, thus implicating 24 different randomizations possibilities. Each subject drew one possibility at random and this possibility was removed from the list to avoid that several subjects get same one. Process was explained at line 96 to 98.
How exactly was VT1 and VT2 determined? Describe the methodology of the determination.
In order to determine VT1 and VT2 graphical presentation of parameter has been displayed with averaging of data every 10 seconds. As agreed by Wasserman and beaver methodology we mainly use VE/VO2 and VE/VCO2 graph to set the position of threshold. VT1 was set at the first increase of VE/VO2 without increase of VE/VCO2 and VT2 was set at concomitant increase of VE/VO2 and VE/VCO2.
You indicate that there was probably a change in tidal volume (VT). Why was this parameter not evaluated?
Vt was collected by device during our experimentation, but we had to limit number of parameters for tables and plots which are already to high. VE being BF x Vt, it was the most appropriate variable to remove from the set.
For clarity of the text and quicker understanding of the results, I recommend to briefly describe in the results section the results found in the tables.
Short introduction of results has been added from line 178 to 184. We cannot add more information without being redundant with other discussion section.
How was the running speed controlled and how was its exact compliance controlled? I recommend the mechanism of implementation, management and control of the pace.
We forget to add this relevant information. Level test was performed using vamEval soundtrack. During uphill test, constant15 seconds pacing was used with dedicated paths with marker at appropriate position and tolerance zone indicated.
Speed control was performed using audio soundtrack read by mobile mp3 player with pacing every 15 seconds. To maintain right speed athlete must be at flag position when sound come out. Test was interrupted if athletes get more than 5 meters difference with appropriate position during 2 successive intervals.
We update document from line 110 to 115.
Line 123: “(60-cm)” – is it correct?
Yes but we update in 0.6 meters as expected by journal recommendations.
I recommend that the wording of the conclusion be reformulated so as not to repeat the literal definition of the project goal.
Right, it was different. We updated the conclusion to be in line with aim of the study claim at the end of the introduction

Round 2
Reviewer 2 Report (Previous Reviewer 1)
All my questions have been answered and comments incorporated. The article is without my further comments.
This manuscript is a resubmission of an earlier submission. The following is a list of the peer review reports and author responses from that submission.
Round 1
Reviewer 1 Report
The main aim of the paper „Physiological Implication of Slope Gradient during Incremen-tal Running Test“ was to to examine the effects of different gradient slopes (from level to +40%) on cardiorespiratory variables reach at exhaustion and ventilatory thresholds during maximum incremental tests in trail runner specialists.
The study is very interesting. I would like to appretiate the efforts of the authors. However, some information is missing in the article, some facts need to be explained:
The methodology section (2.1 Participants) presents the training volume, but does not specify the time period. Per week?
How exactly were the participants and tests randomized?
References in the text are not used in accordance with the rules of the journal.
The list of references is not listed in the order of use in the text.
Table names should be above the table.
Author Response
Dear reviewer,
Thank you for your comments and mistakes pointed out in our document.
We updated the reference list in accordance with the standard of the journal.
You can find below answers to your specific comments.
The methodology section (2.1 Participants) presents the training volume, but does not specify the time period. Per week?
Average training volume for 2 months preceding the study has been implemented in the document. (8.4 ± 3.2 h per week).
How exactly were the participants and tests randomized?
4 different slopes mean 24 possibilities for slopes orders. All possibilities have been listed and then eliminated to avoid similar sequence, as now stated in the revised manuscript.
References in the text are not used in accordance with the rules of the journal.
This has been updated
The list of references is not listed in the order of use in the text.
This has been updated
Table names should be above the table.
Document edition has been done by MDPI. We nevertheless moved the legend in accordance to the comment.
You can see the new version updated based on comments from all reviewers.
Reviewer 2 Report
The paper is potentially interesting and well elaborated from a scientific point of view, however, it has not been sent in the format requested by this journal so that the reviewers can make the indications with reference to the line number in which some modification has to be made.
References in the text have not been respected either, as they have been included according to APA standards and not those indicated by this journal (ACS), as well as the final references.
The legend of the tables is incorrectly included and needs to be revised.
My recommendation is to pay a little attention to the rules and apply them so that it is an evaluable work like any other and to be able to start a correct revision.
Author Response
Thank you for you comments.
Document has been updated with recommendation of journal regarding reference.
The legend of the tables is incorrectly included and needs to be revised.
We nevertheless moved the legend in accordance to the comment.
You can see the new version updated based on comments from all reviewers.
Author Response
Dear Reviewer,
Thank you for your review and comments. We updated citation style according to the rules of MDPI. Edition of the document has been done by MDPI because we did not use the template for submission.
This study is a part of large program aiming to evaluate mountain runners under specific conditions. Protocols designed with positive gradient has been already presented in international congress of Medecine & Science in Ultra-Endurance Sport. 10.1123/ijspp.2018-0227, but not the present study. To design this protocol more than 250 pre-tests have been performed in different trials with different population.
As stated in introduction of the present manuscript, several studies tested incremental test in uphill conditions, but no study investigated slopes with such a magnitude with the specific population of mountain runners. This is clearly mentioned in the last part of the introduction in our manuscript:
Hence, the aim of this study was to examine the effects of different gradient slopes (from level to +40%) on cardiorespiratory variables reach at exhaustion and ventilatory thresholds during maximum incremental tests in trail runner specialists.
- The sample is represented by 10 men and 4 women. Being the behavior in terms of consumption very different due to the obvious physiological differences that exist between sexes such as the amount of hemoglobin. It may also affect the phase of the menstrual cycle in which each of the women is and may show contradictory results.
Yes, that is true, Women are physically different than men, but this is not a problem for the purpose of the study. They remained different than men for the entire study and no comparison between men and women is performed: on the contrary in the present study every subject is its own control (the point is to evaluate the effect of slope).
Menstrual phase is most challenging question because women performed tests in period of 2 weeks. Nevertheless Gordon et al. (10.1111/cpf.12469 ) highlighted that menstrual status impact some cardiorespiratory markers but not VO2Max, cardiac output or maximal heart rate that were not statistically different. Hence a specific effect is unlikely.
- The protocols used need further explanation. It would help to use some image or figure
Thank you, we built and added a new figure to displays all protocols in same line to observe speed and slope evolution.
-On the basis of what scientific evidence were these protocols carried out?
We designed the protocol based on 4 years experiences in doing such tests with elite athletes. Test at level (VamEval) was published more than 20 years ago and test at 25% slope has been published 10.1123/ijspp.2018-0227. As mentioned in the introduction, the specific effect of slope needs to be better defined physiologically speaking and hence the aim of using slope up to 40% in this specific population of trail runners..
- Is there any study that validates the assessment of ventilatory thresholds through Metasoftestudio? If so, it should be cited. How the program calculates ventilatory thresholds should be reported.
As you can read in the document only VO2 Max period was automatically determined by Metasoft studio as the highest average period of 30 second. Ventilatory thresholds were determined by experimented examinator using Wasserman and Beaver Methods. (We do not recommend to use automatic detection).
An experimented examinator determined positions of ventilatory thresholds (VT1) and 2 (VT2) using Wasserman and Beaver methods [36]
Why RER (Respiratory Exchange Ratio; VCO2/VO2) information was not provided? This would provide information about the energy substrate used.
RER is an interesting values, but the aim of this study was to focus on athletes’ evaluation in order to provide training relevant information. RER is not in the top of the list of interested to evaluate athletes.
More information should be provided on the point where the thresholds occurred: speed, test time, RER....
As explain above, we made some choice to stay focus on performance marker. For this reason, Ascending speed in present in the Table 1 and in Figure 2. Time of the test and RER are far away from the topic and are not presented because tables and figures are too big and could not be inserted.
You can see the new version updated based on comments from all reviewers.
Round 2
Reviewer 2 Report
The article is still not sent to the reviewers under the template provided by the journal. This makes it impossible to indicate in which line or lines corrections should be made.
This formal act is the first that the authors should take into account so that we reviewers can do our job and provide them with the necessary corrections.
In the header and footer it appears that the article is from 2020.... Are you sure?
The bibliographic references in the text are still misplaced because the font type changes with respect to the rest of the text.
At the end of the second paragraph of the second page appears "(+0.5 km.h-1 and +1% per minute) and..." when it should be "(+0.5 km-h-1 and +1% per minute) and...". We can find this type of errors frequently throughout the article that due to the lack of a proper template I cannot point out for correction.
It does not include the reference number of approval of the Ethics Committee with which the experiment has been approved. Assuming the principles of the Helsinki Declaration is fine but it is not enough.
The units of measurement sometimes do not appear, for example in the age of the participants, other times it appears next to the value as for example at the beginning of the results (944.0 ± 115.8s, 907.6 ± 99.6s, 900.2± 100.2s and 904.3 ± 101.2s) and other times not respecting the nomenclature of the international system of measurements, kilogram is kg and not Kg.
The end of the third paragraph of the discussion has its last lines with another font "This bigger dispersion ... more pro-nounced at higher gradients.".
And also in the penultimate paragraph of this section "is superior in this gradient. Based on this finding ... and middle-distance trail running competition [57]."
The conclusions are amply improvable after a good article that they insist on not presenting it adequately despite being a good idea, a good methodology and a good statistical treatment.
In summary, once again, the work needs to be done only by the standards of the journal with a little care and not so hastily.
Reviewer 3 Report
The authors have provided more information on the study; however, some important considerations remain:
1. Although I did not put it in the first review, the study mentions that it has been conducted according to the Helsinki ethical statements, however, it specifies that the experimentation must have passed research ethics committee approval. Without this, the study cannot be published as it infringes the laws for experimentation on human subjects. Please provide the ethics committee that evaluated your work and the registration number.
2. I do not agree that test time and RER are not performance factors. How can it not be a performance factor to know if participants use one substrate at a higher rate than another? This is closely related to thresholds
3. Another important aspect is the lack of a control group. The passage of weeks may have affected the performance of the participants. In that case, why was there no control group, or on the other hand, why was there no crossover design?
4. Threshold detection should be cited as well as protocols. Years of experience without scientific substantiation is not supported.
5. Bibliographic citations in the text should be reviewed because they are in a different font format than the rest of the text. This is something that should be reviewed thoroughly, especially after the previous review.
5. As the data are parametric, in many respects comparisons of means work, and this is what is statistically compared, so your statement that the comparison is individual is not valid. In that case a comparison of percent change or Wilcoxon ranks should be used. If so, a table with the individual data for each of the subjects should be provided. In addition, VO2 may not affect maximal exercise, but it does affect the utilization of energy substrates and thus thresholds, so this parameter may affect the results.